# Manipulation of Ascorbate Biosynthetic, Recycling, and Regulatory Pathways for Improved Abiotic Stress Tolerance in Plants

**DOI:** 10.3390/ijms21051790

**Published:** 2020-03-05

**Authors:** Ronan C. Broad, Julien P. Bonneau, Roger P. Hellens, Alexander A.T. Johnson

**Affiliations:** 1School of BioSciences, The University of Melbourne, Melbourne, VIC 3010, Australia; 2Centre for Tropical Crops and Biocommodities, Institute for Future Environments, Queensland University of Technology, Brisbane, QLD 4001, Australia

**Keywords:** ascorbic acid, vitamin c, antioxidant, biosynthesis, recycling, regulation, genetic engineering, genetic modification, genome editing

## Abstract

Abiotic stresses, such as drought, salinity, and extreme temperatures, are major limiting factors in global crop productivity and are predicted to be exacerbated by climate change. The overproduction of reactive oxygen species (ROS) is a common consequence of many abiotic stresses. Ascorbate, also known as vitamin C, is the most abundant water-soluble antioxidant in plant cells and can combat oxidative stress directly as a ROS scavenger, or through the ascorbate–glutathione cycle—a major antioxidant system in plant cells. Engineering crops with enhanced ascorbate concentrations therefore has the potential to promote broad abiotic stress tolerance. Three distinct strategies have been utilized to increase ascorbate concentrations in plants: (i) increased biosynthesis, (ii) enhanced recycling, or (iii) modulating regulatory factors. Here, we review the genetic pathways underlying ascorbate biosynthesis, recycling, and regulation in plants, including a summary of all metabolic engineering strategies utilized to date to increase ascorbate concentrations in model and crop species. We then highlight transgene-free strategies utilizing genome editing tools to increase ascorbate concentrations in crops, such as editing the highly conserved upstream open reading frame that controls translation of the *GDP-L-galactose phosphorylase* gene.

## 1. Introduction

Providing adequate, safe, and nutritious food for the growing human population under changing climatic conditions represents one of the greatest challenges of the 21^st^ century. Adverse environmental conditions such as extreme temperatures, drought, and soil salinity are the primary cause of crop losses worldwide and are predicted to be exacerbated by climate change [1]. With the human population expected to reach 9.7 billion by the year 2050, the need for more abiotic stress tolerant crops is critical for ensuring global food security [2].

A common consequence of many abiotic stresses is the overproduction of reactive oxygen species (ROS) such as singlet oxygen (^1^O_2_), superoxide (O_2_^−^), hydrogen peroxide (H_2_O_2_), and the hydroxyl radical (HO^−^) [3]. For example, stomatal closure during drought and salt stress limits carbon dioxide uptake, which has both direct and indirect effects on ROS production. For instance, reduced NADP^+^ regeneration through the Calvin cycle or enhanced photorespiration results in a higher leakage of photosynthetic or respiratory electrons, that in turn stimulate ROS production [4,5]. Similarly, Na^+^ and Cl^-^ toxicity during salt stress can disrupt the photosynthetic electron transport chain, further promoting electron leakage and ROS accumulation [6]. Likewise, temperature stresses are capable of uncoupling temperature-sensitive pathways, such as photosynthesis and respiration, consequently increasing ROS formation [7]. Identifying mechanisms to enhance the capacity of plants to mitigate the harmful effects of the overproduction of ROS will greatly facilitate the production of crops with broad abiotic stress tolerance.

Ascorbate, also known as vitamin C, is an important multifunctional molecule for both plants and animals. Ascorbate is a reducing agent capable of donating electrons and primarily functions as a cellular antioxidant and enzymatic co-factor. In plants, ascorbate is the most abundant water-soluble antioxidant and is found in high concentrations in peroxisomes (22.8 mM), the cytosol (21.7 mM), and nuclei (16.3 mM), intermediate concentrations in mitochondria (10.4 mM) and chloroplasts (10.8 mM), low concentrations in the vacuole (2.3 mM), and very low concentrations in the apoplast [8,9]. Ascorbate is particularly well-known for its roles in photosynthetic functions and stress tolerance. This is largely due to its ability to counteract oxidative stress produced by normal or stressed cellular metabolism, either directly as a ROS scavenger or through the ascorbate-glutathione cycle—a major antioxidant system of plants cells (Figure 1) [10,11,12]. In this cycle, H_2_O_2_ is detoxified to H_2_O by ascorbate peroxidase (APX), using ascorbate as an electron donor. The oxidized ascorbate can then be regenerated by the ascorbate recycling enzymes monodehydroascorbate reductase (MDAR) or dehydroascorbate reductase (DHAR) using NADPH and glutathione, respectively, as electron donors (see Section 4) [3,11].

To date, four ascorbate biosynthetic pathways have been proposed in plants: the L-galactose, L-gulose, *myo*-inositol, and D-galacturonate pathways [13,14]. The L-galactose pathway, however, is the only pathway to have all the enzymatic steps characterized. Substantial genetic evidence now also supports the L-galactose pathway as the dominant pathway towards ascorbate biosynthesis in plants. Increased expression of ascorbate biosynthetic genes—particularly L-galactose pathway genes—has proven to be an effective strategy to significantly increase ascorbate concentrations in plants. In addition to de novo biosynthesis, cellular ascorbate concentrations are also maintained via the ascorbate recycling enzymes MDAR and DHAR. Increased expression of these ascorbate recycling genes has also proven to be an effective strategy for increasing the concentrations of reduced, active ascorbate in plants. While the genetic pathways underlying the biosynthesis and recycling of ascorbate have largely been understood for over a decade, the genetic pathways underlying the regulation of ascorbate biosynthesis are only now starting to be unraveled. Manipulating ascorbate regulatory factors is emerging as a promising new strategy to increase ascorbate concentrations in plants.

Here, we review the genetic pathways underlying ascorbate biosynthesis, recycling, and regulation in plants, including a summary of all metabolic engineering strategies utilized to date to increase ascorbate concentrations in model and crop species.

## 2. Biosynthesis of Ascorbate in Plants

Four pathways towards ascorbate biosynthesis have been proposed in plants: the L-galactose, L-gulose, *myo*-inositol, and D-galacturonate pathways [13,14] (Figure 2). All four pathways share an aldonolactone as the direct precursor to ascorbate (L-galactono-1,4-lactone for the L-galactose and D-galacturonate pathways and L-gulono-1,4-lactone for the L-gulose and *myo*-inositol pathways). While there is strong evidence for the L-galactose pathway as the dominant ascorbate biosynthetic pathway in plants, the contribution of the L-gulose, *myo*-inositol, and D-galacturonate pathways towards ascorbate biosynthesis in plants is not as well documented and often controversial. An overview of metabolic engineering strategies utilizing ascorbate biosynthetic genes to increase ascorbate concentrations in model and crop species, including the tissue examined and any stress tolerances observed, is summarized in Table 1.

### 2.1. The L-Galactose Pathway

The L-galactose pathway (also known as the D-mannose/L-galactose pathway or Smirnoff-Wheeler pathway) is the only ascorbate biosynthetic pathway in plants to have all its enzymatic steps characterized. The L-galactose pathway is responsible for converting D-fructose-6-P to ascorbate via eight enzymatic steps by phosphomannose isomerase (PMI), phosphomannose mutase (PMM), GDP-D-mannose pyrophosphorylase (GMP), GDP-D-mannose-3′,5′-epimerase (GME), GDP-L-galactose phosphorylase (GGP), L-galactose-1-phosphate phosphatase (GPP), L-galactose dehydrogenase (L-GalDH), and L-galactono-1,4-lactone dehydrogenase (L-GalLDH) [13,14] (Figure 2). All enzymatic steps of the L-galactose pathway take place in the cytosol, except for the conversion of L-galactono-1,4-lactone to ascorbate by L-GalLDH, which occurs in mitochondria [15].

#### 2.1.1. Phosphomannose Isomerase (PMI)

The PMI enzyme is responsible for the first enzymatic step of the L-galactose pathway, catalyzing a reversible conversion of D-fructose-6-P to D-mannose-6-P (Figure 2). Knock-down of the *PMI1* gene in *Arabidopsis thaliana* by RNAi reduced ascorbate to 47% of wild-type levels, supporting its role in L-galactose ascorbate biosynthesis [16]. To date, no studies have investigated the stable increased expression of the *PMI* gene in plants. However, transient expression of the *PMI1* gene in *Arabidopsis* did not significantly change ascorbate concentrations, indicating that PMI is not a rate-limiting step in the L-galactose pathway in *Arabidopsis* [17].

#### 2.1.2. Phosphomannose Mutase (PMM)

The PMM enzyme is responsible for the second enzymatic step of the L-galactose pathway, catalyzing a reversible conversion of D-mannose-6-P to D-mannose-1-P (Figure 2). Temperature-sensitive mutations in the *Arabidopsis PMM* gene reduced ascorbate to 20% of wild-type levels, supporting its important role in L-galactose ascorbate biosynthesis [18]. Increased expression of the *PMM* gene has increased ascorbate concentrations 1.3-fold in *Arabidopsis* [19] and 2.5-fold in tobacco (*Nicotiana tabacum* L.) [20] (Table 1). The small increase in ascorbate concentrations observed in the *Arabidopsis* plants with increased expression of the *PMM* gene was not proportionate to the highly upregulated PMM enzymatic activity, indicating that PMM is not a rate-limiting step of the L-galactose pathway in *Arabidopsis* [19]. Increased expression of the *PMM* gene in *Arabidopsis* was associated with enhanced tolerance to methyl viologen-induced stress [19] (Table 1).

#### 2.1.3. GDP-D-Mannose Pyrophosphorylase (GMP)

The GMP (also known as VITAMIN C-1 [VTC1]) enzyme is responsible for the third enzymatic step of the L-galactose pathway, catalyzing a reversible conversion of D-mannose-1-P to GDP-D-mannose (Figure 2). Missense mutations in the *Arabidopsis GMP* gene reduced ascorbate to 25% of wild-type levels, supporting its important role in L-galactose ascorbate biosynthesis [21,22,23]. Increased expression of the *GMP* gene has increased ascorbate concentrations in a range of species, including 1.3-fold in *Arabidopsis* [24], 2.5-fold in tobacco [25], 1.4- to 1.7-fold in tomato (*Solanum lycopersicum* L.) [26,27], and 1.5-fold in rice [28] (Table 1). Increased co-expression of the *GMP* + *GME* genes in tomato increased ascorbate concentrations 2.0-fold, a greater increase than the individual genes alone, suggesting that GMP may act synergistically with GME to increase ascorbate concentrations in tomato [26]. Increased co-expression of the *GMP* + *GME* + *GGP* + *GPP* genes in tomato also increased ascorbate concentrations 2.0-fold; however, this was the same fold-change for increased expression of the *GGP* gene alone, suggesting that GMP is not a major rate-limiting step in the L-galactose pathway in tomato [26]. This conclusion is also supported by other studies increasing the expression of the *GMP* gene in *Arabidopsis* and tobacco and increasing the co-expression of the *GMP* + *GME* genes in tobacco that did not report significant changes in ascorbate concentrations [29,30] (Table 1).

#### 2.1.4. GDP-D-Mannose-3′,5′-Epimerase (GME)

The GME enzyme is responsible for the fourth enzymatic step of the L-galactose pathway, catalyzing a reversible conversion of GDP-D-mannose to produce GDP-L-galactose (Figure 2). The GME enzyme also catalyzes a reversible conversion of GDP-D-mannose to produce GDP-L-gulose. Consequently, an alternative L-gulose pathway has been proposed to contribute towards ascorbate biosynthesis in plants (see Section 2.2) [31]. Knock-down of the *GME* gene in tomato by RNAi reduced ascorbate to 44% of wild-type levels, supporting its important role in L-galactose ascorbate biosynthesis [32]. Increased expression of the *GME* gene has increased ascorbate concentrations 1.4- to 1.9-fold in *Arabidopsis* [24,33,34], 1.4- to 1.8-fold in tomato [26,35], and 1.9-fold in rice [28]. Increased expression of the *GMP* + *GME* + *GGP* + *GPP* genes in tomato only increased ascorbate concentrations 2.0-fold, the same fold-change for increased expression of the *GGP* gene alone, suggesting that GME is not a major rate-limiting step in the L-galactose pathway in tomato [26]. Moreover, increased expression of the *GME* gene in tobacco did not significantly change ascorbate concentrations, suggesting the GME is not a rate-limiting step of the L-galactose pathway in tobacco [29]. Increased expression of the *GME* gene in plants has been associated with enhanced tolerance to salt, low pH, drought, cold, and methyl viologen-induced stress [28,33,35] (Table 1).

#### 2.1.5. GDP-L-Galactose Phosphorylase (GGP)

The GGP (also known as VTC2/5) enzyme is responsible for the fifth enzymatic step of the L-galactose pathway, catalyzing the conversion of GDP-L-galactose to L-galactose1-P and represents the first committed step towards ascorbate biosynthesis, since GDP-D-mannose and GDP-L-galactose are also used as precursors for cell wall polysaccharides and glycoproteins (Figure 2). Splice junction and missense mutations in the *Arabidopsis GGP* gene reduced ascorbate to 10% of wild-type levels, supporting its critical role in L-galactose ascorbate biosynthesis [36,37,38,39]. The *ggp1*(*vtc2*)/*ggp2*(*vtc5*) double mutants in *Arabidopsis* also exhibit a seedling-lethal phenotype without the supplementation of L-galactose or ascorbate [38,39]. These results support the pivotal role of GGP in L-galactose ascorbate biosynthesis, as well as highlight the importance of L-galactose ascorbate biosynthesis for plant growth and development. Increased expression of the *GGP* gene has increased ascorbate concentrations in a wide range of species, including 2.9- to 4.1-fold in *Arabidopsis* [24,40], 3.1-fold in potato (*Solanum tuberosum* L.) [41], 2.1-fold in strawberry (*Fragaria* × *ananassa*) [41], 2.0- to 6.2-fold in tomato [26,41], 1.4-fold in tobacco [42], and 2.5- to 2.6-fold in rice [28,43] (Table 1). Increased co-expression of the *GGP* + *L-GalLDH* genes or the *GGP* + *GPP* genes increased ascorbate concentrations 3.6- and 4.1-fold, respectively, in *Arabidopsis*, a greater fold-change than increased expression of the *GGP* gene alone, suggesting that GGP may act synergistically with L-GalLDH and GPP in the L-galactose pathway [24]. However, increased co-expression of the *GGP* + *GPP* genes in tomato only increased ascorbate concentrations 1.8-fold, less than the fold-change for increased expression of the *GGP* gene alone, suggesting that GGP and GPP do not act synergistically in the L-galactose pathway in tomato [26]. Relative to other genes from the L-galactose pathway, increased expression of the *GGP* gene consistently leads to the greatest increase in ascorbate concentrations, providing strong genetic evidence that GGP represents the rate-limiting enzymatic step of the L-galactose pathway in many species [17,24,26,28,40]. Consistent with being the first committed step towards ascorbate biosynthesis and rate-limiting step of the L-galactose pathway, the strongest regulation of ascorbate concentrations in plants has been proposed to occur through the *GGP* gene [44]. Increased expression of the *GGP* gene in plants has been associated with enhanced tolerance to salt, ozone, and cold stress [28,42,43] (Table 1).

#### 2.1.6. L-Galactose-1-Phosphate Phosphatase (GPP)

The GPP (also known as VTC4) enzyme is responsible for the sixth enzymatic step of the L-galactose pathway, catalyzing the conversion of L-galactose1-P to L-galactose (Figure 2). A missense mutation in the *Arabidopsis GPP* gene reduced ascorbate to 50% of wild-type levels, supporting its important role in L-galactose ascorbate biosynthesis [36,45]. Increased expression of the *GPP* gene has increased ascorbate concentrations 1.5-fold in *Arabidopsis* [24], 1.7-fold in tomato [26], and 1.4-fold in rice [28] (Table 1). Increased co-expression of the *GGP* + *GPP* genes or the *GMP* + *GME* + *GGP* + *GPP* genes in tomato did not increase ascorbate concentrations higher than increased expression of the *GGP* gene alone, indicating that GPP is not a major rate-limiting step in the L-galactose pathway in tomato [26].

#### 2.1.7. L-Galactose Dehydrogenase (L-GalDH)

The L-GalDH enzyme is responsible for the penultimate enzymatic step of the L-galactose pathway, catalyzing the conversion of L-galactose to L-galactono-1,4-lactone (Figure 2). Knock-down of the *L-GalDH* gene in *Arabidopsis* with RNAi had no effect on ascorbate concentrations under low light conditions, but reduced ascorbate to 45% of wild-type levels under high light conditions when a larger pool of ascorbate is required, supporting its role in L-galactose ascorbate biosynthesis [46]. Increased expression of the *L-GalDH* gene increased ascorbate concentrations 1.2-fold in *Arabidopsis* [24] and 1.7-fold in rice [28] (Table 1). However, increased expression of the *L-GalDH* gene in tobacco did not significantly change ascorbate concentrations, despite having a 3.5-fold increase in L-GalDH enzymatic activity, indicating that L-GalDH is not a rate-limiting step in the L-galactose pathway in tobacco [46].

#### 2.1.8. L-Galactono-1,4-Lactone Dehydrogenase (L-GalLDH)

The L-GalLDH enzyme is responsible for the final enzymatic step of the L-galactose pathway, catalyzing the conversion of L-galactono-1,4-lactone to ascorbate (Figure 2). Similar to the *ggp1*(*vtc2*)/*ggp2*(*vtc5*) double mutants in *Arabidopsis*, loss-of-function *Arabidopsis l-galldh* mutants exhibited a seedling lethal phenotype in the absence of ascorbate supplementation, supporting the critical role of L-GalLDH in L-galactose ascorbate biosynthesis [47]. Increased expression of the *L-GalLDH* gene has increased ascorbate concentrations in a range of species, including 1.8-fold in *Arabidopsis* [24], 1.3- to 3.2-fold in lettuce (*Lactuca sativa* L.) [48,49], 7.0-fold in lily (*Lilium davidii*) [50], 1.3- to 1.5-fold in rice [28,51], and 2.1-fold in tobacco [52] (Table 1). One study, however, found that increased expression of the *L-GalLDH* gene in tobacco did not significantly change ascorbate concentrations, despite having 10-fold higher L-GalLDH enzymatic activity, suggesting that L-GalLDH is not a major rate-limiting step of the L-galactose pathway in tobacco [53]. Increased co-expression of the *GGP* + *L-GalLDH* genes increased ascorbate concentrations 3.6-fold in *Arabidopsis*, a greater fold-change than increased expression of each gene alone, suggesting that L-GalLDH may act synergistically with GGP in the L-galactose pathway in *Arabidopsis* [24]. Increased expression of the *L-GalLDH* gene in tobacco was associated with enhanced tolerance to salt and methyl viologen-induced stress [52] (Table 1).

### 2.2. The L-Gulose Pathway

The L-gulose pathway (also known as the L-gulose shunt) is a result of the epimerization of GDP-D-mannose to GDP-L-gulose by GME, in addition to the epimerization of GDP-D-mannose to GDP-L-galactose [31] (Figure 2). In this pathway, GDP-L-gulose is proposed to be converted to the ascorbate precursor L-gulono-1,4-lactone via L-gulose-1-P and L-gulose intermediates; however, the genes and enzymes responsible for catalyzing these steps in plants have not yet been identified. The GGP and L-GalDH enzymes from the L-galactose pathway have been proposed to catalyze these reactions; however, GGP does not appear to have any enzymatic activity on GDP-L-gulose and L-GalDH has a much lower affinity for L-gulose than L-galactose [46,54,55]. L-gulono-1,4-lactone is then proposed to be converted to ascorbate by L-gulono-1,4-lactone oxidase (L-GulLO) (Figure 2). Increased expression of the rat (*Rattus rattus* L.) *L-GulLO* gene has increased ascorbate concentrations in a range of species, including 1.8- to 2.0-fold in *Arabidopsis* [56,57], 7.0-fold in lettuce [58], 2.4-fold in potato [59], 7.0-fold in tobacco [58], and 1.5-fold in tomato [60] (Table 1). However, increased expression of the *Arabidopsis L-GulLOs* genes in *Arabidopsis* did not change ascorbate concentrations [61] (Table 1) and increased expression of the *Arabidopsis L-**GulLOs* genes in tobacco cells only increased ascorbate concentrations when supplied with L-gulono-1,4-lactone, together suggesting that the L-gulose pathway is not a major contributor to ascorbate biosynthesis in plants [62]. Increased expression of the rat *L-GulLO* gene in plants has been associated with enhanced tolerance to salt, cold, heat, drought, and pyrene- and methyl viologen-induced stress [56,59,60] (Table 1).

### 2.3. The Myo-Inositol Pathway

The *myo*-inositol pathway (also known as the D-gluconorate pathway), which operates in animals, has also been proposed to be present in plants (Figure 2). In this pathway, *myo*-inositol is proposed to be converted to the ascorbate precursor L-gulono-1,4-lactone via D-gluconorate and D-gulonate intermediates; however, the gene and enzyme responsible for catalyzing the conversion of D-gluconorate to D-gulonate have not yet been identified (Figure 2). L-gulono-1,4-lactone is then proposed to be converted to ascorbate by L-GulLO (Figure 2). The contribution of the *myo*-inositol pathway towards ascorbate biosynthesis in plants has been controversial. Supporting its contribution are studies that increased the expression of the *myo-inositol oxygenase* (*MIOX*) gene, which encodes the enzyme responsible for catalyzing the conversion of *myo*-inositol to D-glucuronate (Figure 2). For example, increased expression of the *MIOX* gene in *Arabidopsis* increased ascorbate concentrations 1.6- to 3.0-fold, and was associated with enhanced tolerance to salt, cold, heat, and pyrene-induced stress [56,63,64] (Table 1). Varying results were obtained when the expression of the *MIOX* gene was increased in tomato, with ascorbate reduced to 50% of wild-type levels in leaves, but increased 1.4- and 1.3-fold in green and red fruit, respectively, indicating that the *myo*-inositol pathway may be tissue-specific in tomato [27]. Further support for the *myo*-inositol pathway comes from increased expression of phytases that are proposed to increase the pool of *myo*-inositol, which can then serve as an entry point for ascorbate biosynthesis via the *myo*-inositol pathway. For example, increased expression of the *Arabidopsis* purple acid phosphatase 15 (*PAP15*) gene [65] and the bacterial beta-propeller phytase PHY-US417 gene [66] in *Arabidopsis* has increased ascorbate concentrations 2.0- to 2.3-fold and were associated with enhanced tolerance to drought and salt stress. Loss-of-function mutations in the *Arabidopsis PAP15* gene reduced ascorbate to 70% of wild-type levels, supporting the contribution of PAP15 towards *myo*-inositol ascorbate biosynthesis [65]. The *PAP15* gene, however, has also been proposed to encode a GPP in *Arabidopsis*, which may provide an alternative explanation for these results [13]. While there are several studies supporting the contribution of the *myo*-inositol pathway towards ascorbate biosynthesis in plants, there are equally as many studies opposing its contribution. For example, increased expression of the *MIOX* gene in *Arabidopsis* [67] and rice [68] did not change ascorbate concentrations, despite the *Arabidopsis* lines with increased *MIOX* expression having higher MIOX enzymatic activity (Table 1). Further, loss-of-function mutations in the *Arabidopsis* [69] and rice [70] *MIOX* genes did not change ascorbate concentrations relative to wild-type, challenging the view that the *myo*-inositol pathway contributes towards ascorbate biosynthesis. Further opposition comes from a recent study utilizing CRISPR/Cas9-induced knockout mutations in the *Arabidopsis glucuronokinase1* (*GlcAK1*) gene and radiotracer experiments with ^3^H-*myo*-inositol. Knockout of *GlcAK1* ought to increase the flux from *myo*-inositol to ascorbate if the *myo*-inositol pathway is operational; however, no changes in ascorbate concentrations were detected in the *Arabidopsis glcak1* mutants. Moreover, radiolabeled ^3^H-*myo*-inositol in the *Arabidopsis glcak1* mutants accumulated as D-gluconorate and ascorbate remained unlabeled, leading the authors to conclude that the *myo*-inositol pathway does not contribute towards ascorbate biosynthesis in plants [71].

### 2.4. The D-Galacturonate Pathway

Evidence for the D-galacturonate pathway was discovered in strawberry fruits [72]. In this pathway, D-galacturonate derived from pectin in the cell wall is proposed to be converted to the ascorbate precursor L-galactono-1,4-lactone via a L-galactonate intermediate; however, the gene and enzyme responsible for converting L-galactonate to L-galactono-1,4-lactone have not yet been identified in plants (Figure 2). L-galactono-1,4-lactone is then proposed to be converted to ascorbate by L-GalLDH, the final enzymatic step of the L-galactose pathway (Figure 2). To date, the *D-galacturonate reductase* (*D-**GalUR*) gene, which encodes the enzyme responsible for catalyzing the conversion of D-galacturonate to L-galactonate, has only been identified in strawberry (Figure 2). Knock-out or knock-down studies of the *D-**GalUR* gene in strawberry or other plant species have not yet been reported, making it difficult to determine the relative contribution of the D-galacturonate pathway towards ascorbate biosynthesis in plants. Nevertheless, increased expression of the *D-**GalUR* gene has increased ascorbate concentrations 3.0-fold in *Arabidopsis* [72], 1.7- to 2.1-fold in potato [73,74,75], and 1.0- to 2.5-fold in tomato [76,77,78]. Increased expression of the *D-**GalUR* gene in plants has been associated with enhanced tolerance to salt, drought, cold, zinc chloride, and methyl viologen-induced stress [73,74,75,77,78] (Table 1).

## 3. Recycling of Ascorbate in Plants

In addition to de novo biosynthesis of ascorbate, oxidized ascorbate can be regenerated via the recycling enzymes MDAR and DHAR to maintain cellular ascorbate levels (Figure 1). Following ascorbate oxidation, a short-lived monodehydroascorbate radical is produced. Monodehydroascorbate can then be recycled back to ascorbate by MDAR using NADPH as a reductant, however if not recycled monodehydroascorbate will rapidly undergo non-enzymatic disproportionation with another monodehydroascorbate molecule to produce an ascorbate and dehydroascorbate molecule [81]. Dehydroascorbate can then be recycled back to ascorbate by DHAR using glutathione as a reductant, however if not recycled dehydroascorbate will undergo an irreversible hydrolysis to generate 2,3-diketogulonic acid [81]. An overview of metabolic engineering strategies utilizing ascorbate recycling genes to increase ascorbate concentrations in model and crop species, including the tissue examined and any stress tolerances observed, is summarized in Table 2.

Loss-of-function mutations in the *Arabidopsis MDAR4* gene reduced the ratio of ascorbate to dehydroascorbate but did not affect the pool size of total ascorbate (ascorbate + dehydroascorbate), supporting the role of MDAR in ascorbate recycling [82]. Increased expression of the *MDAR* gene has increased the concentrations of reduced ascorbate 1.2- to 2.2-fold in tobacco [83,84,85] and 1.2-fold in tomato [86] (Table 2). However, other studies increasing the expression of the *MDAR* gene in tobacco [87] and tomato [88] did not report any significant changes to reduced ascorbate concentrations, indicating that MDAR may not be a rate-limiting step in the recycling of ascorbate in these species (Table 2). Increased expression of the *MDAR* gene in plants has been associated with enhanced tolerance to salt, ozone, drought, cold, heat, and methyl viologen-induced stress [83,84,86,87] (Table 2).

Similar to the *mdar4* mutant in *Arabidopsis*, loss-of-function mutations in the *Arabidopsis DHAR3* gene reduced the ratio of ascorbate to dehydroascorbate but did not affect the pool size of total ascorbate, supporting the role of DHAR in ascorbate recycling [89]. Increased expression of the *DHAR* gene has increased the concentrations of reduced ascorbate in a wide range of species, including 1.1- to 3.3-fold in *Arabidopsis* [34,90,91,92,93,94], 1.9- to 6.1-fold in maize (*Zea mays* L.) [95,96], 1.3- to 2.8-fold in potato [97,98,99], 1.7-fold in rice [100], 1.3- to 3.9-fold in tobacco [85,95,101,102,103,104,105,106], and 1.5-fold to 1.9-fold in tomato [88,107,108] (Table 2). Increased expression of the *DHAR* gene in plants has been associated with enhanced tolerance to salt, drought, high light, heat, ozone, cold, aluminum, hydrogen peroxide, and methyl viologen-induced stress [85,90,91,92,93,94,98,101,102,103,104,105,106,107,108] (Table 2).

## 4. Regulation of Ascorbate Biosynthesis in Plants

The genes and enzymes responsible for the major biosynthetic route towards ascorbate biosynthesis—the L-galactose pathway—and the recycling of ascorbate in plants have been well characterized for over a decade now [37,55]. The regulation of ascorbate biosynthesis and recycling, on the other hand, has remained relatively unexplored until more recently. Several key regulators have now been identified, and it is becoming clear that ascorbate biosynthesis and recycling is tightly regulated at the transcriptional, translational, and post-translational level [44]. An overview of metabolic engineering strategies utilizing ascorbate regulatory factors to increase ascorbate concentrations in model and crop species, including the tissue examined and any stress tolerances observed, is summarized in Table 3.

### 4.1. Ascorbic Acid Mannose Pathway Regulator 1 (AMR1)

The *ascorbic acid mannose pathway regulator 1* (*AMR1*) gene was identified in an activation-tagged *Arabidopsis* mutant that contained 40% of wild-type ascorbate levels [109]. The *AMR1* gene encodes an F-box protein that negatively regulates the transcription of genes of the L-galactose pathway, including *GMP*, *GME*, *GGP1*, *GPP*, *L-GalDH*, and *L-GalLDH* [109]. The expression of the *AMR1* gene was developmentally and environmentally controlled with increased expression as the plants aged and decreased expression under high light [109]. Loss-of-function mutations in the *AMR1* gene in *Arabidopsis* increased ascorbate concentrations 2.0-fold and were associated with enhanced ozone stress [109] (Table 3). Disrupting the function of the *AMR1* gene with genome editing tools presents one possible transgene-free strategy to increase ascorbate concentrations in crops.

### 4.2. Basic Helix-Loop-Helix 59 Transcription Factor (bHLH59)

The tomato *basic helix-loop-helix*
*59* (*bHLH59*) gene was recently identified as the likely candidate underlying the ascorbate quantitative trait locus TFA9 in tomato [110]. The bHLH59 transcription factor positively regulates the transcription of genes of the L-galactose pathway, including *PMI*, *PMM*, *GMP1*, *GMP2*, *GMP3*, *GMP4*, and *GME1* [110]. Knock-down of the *bHLH59* gene with RNAi in tomato reduced ascorbate to 65% of wild-type levels [110]. Increased expression of the *bHLH59* gene in tomato increased ascorbate concentrations 1.5-fold and was associated with enhanced tolerance to methyl viologen-induced stress [110] (Table 3).

### 4.3. Calmodulin-Like 10 (CML10)

A calcium sensor, calmodulin-like 10 (CML10), was recently identified to interact with *Arabidopsis* PMM in a yeast two-hybrid screen [111]. The expression of the *CML10* gene was upregulated under oxidative stress and the CML10 protein increased PMM enzymatic activity in a calcium-dependent manner [111]. Knock-down of the *CML10* gene in *Arabidopsis* by RNAi reduced ascorbate to 70% of wild-type levels and was associated with decreased drought and H_2_O_2_-induced stress [111]. To date, no studies have investigated increased expression of the *CML10* gene in plants.

### 4.4. COP9 Signalosome Subunit 5B and 8 (CSN5B and CSN8)

The *Arabidopsis* COP9 signalosome subunit 5B (CSN5B), a component of the photomorphogenic COP9 signalosome, was identified to interact with GMP in a yeast two-hybrid screen [112]. Under darkness, CSN5B promotes the ubiquitination and degradation of GMP through the 26S proteasome pathway [112]. Loss-of-function mutations in the *CSN5B* gene in *Arabidopsis* increased ascorbate concentrations 1.4-fold and were associated with enhanced salt stress [112] (Table 3). Loss-of-function mutations in the *CSN8* gene in *Arabidopsis* also increased ascorbate concentrations 1.8-fold (Table 3); however, CSN8 did not interact with GMP in the yeast-two hybrid screen and its mechanism of action has yet to be determined [112]. Disrupting the function of the *CSN5B* or *CSN8* genes with genome editing tools presents another transgene-free strategy to increase ascorbate concentrations in crops.

### 4.5. DNA-Binding with One Finger 22 (Dof22)

The DNA-binding with one finger 22 (Dof22), a member of the Dof family of plant specific transcription factors, has recently been proposed as a negative transcriptional regulator of L-galactose pathway, ascorbate recycling, and ascorbate-glutathione cycle genes in tomato, such as *GGP1*, *GGP2*, *L-GalDH*, *L-GalLDH*, *MDAR*, cytosolic *APX*, and *glutathione reductase* (*GR*) [113]. Knock-down of the *Dof22* gene with RNAi in tomato increased ascorbate concentrations up to 1.6-fold (Table 3), but was associated with reduced tolerance to salt stress due to the transcriptional downregulation of the *salt overly sensitive 1* (*SOS1*) gene, which encodes a plasma membrane Na^+^/H^+^ antiporter [113].

### 4.6. Ethylene Response Factor 98 (ERF98)

In a reverse genetic screen of a pool of *Arabidopsis ethylene response factor* (*ERF*) T-DNA insertion mutants, loss-of-function mutations in the *ERF98* gene reduced ascorbate to 65% of wild-type levels [114]. The *ERF98* gene encodes a transcription factor belonging to the AP2/ERF superfamily that positively regulates the transcription of ascorbate biosynthetic, ascorbate recycling, and ascorbate-glutathione cycle genes, including *GMP*, *GGP1*, *L-GalDH*, *L-GalLDH*, *MIOX4*, *MDAR3*, chloroplastic *DHAR*, cytosolic *DHAR*, and *GR1* [114]. Increased expression of the *ERF98* gene in *Arabidopsis* increased ascorbate concentrations 1.7-fold and was associated with enhanced tolerance to salt stress [114] (Table 3).

### 4.7. GGP Upstream Open Reading Frame (GGP uORF)

The translation of *GGP* has been discovered to be post-transcriptionally regulated through a highly conserved, *cis*-acting upstream open reading frame (uORF) in the long 5’ leader sequence of the *GGP* mRNA [115,116]. The *GGP* uORF is proposed to initiate from a non-canonical AUC or ACG start-codon and encode a 60- to 65-residue long peptide [115]. Disruption of the *GGP* uORF increased ascorbate concentrations when a *GGP* promoter-*uORF*-*GGP* construct was transiently transformed in *Nicotiana benthamiana* [115]. It is proposed that under high ascorbate concentrations, the *GGP* uORF is translated and causes ribosomal stalling, thereby preventing translation of the *GGP* major ORF (mORF), while under low ascorbate concentrations, the uORF is skipped and the *GGP* mORF is translated [115]. The precise mechanism of how ascorbate influences the translation of the *GGP* uORF or mORF, however, is yet to be determined. Recently, disruption of the *GGP* uORF with the CRISPR/Cas9 genome editing system increased ascorbate concentrations 1.7-fold in *Arabidopsis* [117], 1.4- to 2.6-fold in lettuce [117], and 1.4-fold in tomato [118] (Table 3). Moreover, disruption of the lettuce *GGP* uORFs was associated with enhanced tolerance to methyl viologen-induced stress [117] (Table 3). Disrupting the *GGP* uORF with the CRISPR/Cas9 genome editing system has established itself as a very promising transgene-free strategy to engineer plants with enhanced production of GGP for increased ascorbate biosynthesis.

### 4.8. HD-Zip I Family Transcription Factor 24 (HZ24)

The HD-Zip I family transcription factor 24 (HZ24) was recently identified based on a yeast-one hybrid assay to the promoter region of tomato *GMP3* [119]. Knock-down mutants of the *HZ24* gene with RNAi in tomato reduced ascorbate to 60% of wild-type levels [119]. The HZ24 transcription factor positively regulates the transcription of a wide range of genes from the L-galactose pathway in tomato, including *PMM, GMP4*, *GME1, GME2*, *GGP, GPP1, GPP2*, and *L-GalDH* [119]. Increased expression of the *HZ24* gene in tomato increased ascorbate concentrations 1.5-fold and was associated with enhanced tolerance to methyl viologen-induced stress [119] (Table 3).

### 4.9. High-Pigment-1 (HP1)

The *High-Pigment-1* (*HP1*) tomato mutants—previously characterized as having increased carotenoid and flavonoid concentrations—were found to have reduced ascorbate concentrations at various stages of fruit development and ripening relative to wild-type [120]. The *HP1* gene is orthologous to the *Arabidopsis UV-DAMAGED DNA-BINDING PROTEIN 1* (*DDB1*) gene which encodes a component of the CUL4-based E3 ligase complex [121]. Fourteen genes involved in ascorbate biosynthesis and recycling were differentially expressed in the developing fruit of the *hp1* mutant and it was proposed that HP1 positively regulates the transcription of the *GMP* and *GME1* genes, but negatively regulates the transcription of the *L-GalLDH* gene in tomato, ultimately resulting in the reduced ascorbate concentrations observed in the developing and ripening fruit [120]. To date, no studies have investigated the effect of increasing the expression of the *HP1* gene has on ascorbate concentrations in plants.

### 4.10. KONJAC 1 and 2 (KJC1 and KJC2)

Two nucleotide sugar phosphorylase-like proteins, KONJAC 1 and 2 (KJC1 and KJC2), were identified based on their phylogenetic relationship to GMP [30]. Loss-of-function mutations in the *KJC1* and *KJC2* genes in *Arabidopsis* reduced ascorbate to 40% and 70% of wild-type levels, respectively, due to reducing GMP enzymatic activity to 10% and 70% of wild-type levels, respectively [30]. The KJC1 and KJC2 proteins are proposed to post-translationally regulate GMP enzymatic activity and may function opposite that of the COP9 signalosome in the regulation of GDP-D-mannose biosynthesis [30]. Increased expression of the *KJC1* gene increased ascorbate concentrations 1.4-fold in *Arabidopsis* but increased co-expression of the *KJC1* and *GMP* genes or increased expression of the *KJC2* gene did not change ascorbate concentrations [30] (Table 3).

### 4.11. Myeloblastosis Transcription Factor 5 (MYB5)

Pear (*Pyrus betulaefolia*) myeloblastosis transcription factor 5 (MYB5) was recently identified in a yeast one-hybrid screen to interact with the promoter region of the *DHAR2* gene [122]. The expression of the *MYB5* gene was induced by a wide range of stresses, including cold, drought, and salt stress [122]. The MYB5 protein positively regulates the transcription of ascorbate recycling, ascorbate-glutathione cycle, and stress-associated genes, including *MDAR*, *DHAR2*, *APX*, *cystathionine-β-synthase 1* (*CBF1*), *CBF2*, and *CBF3* [122]. Increased expression of the *MYB5* gene in tobacco increased ascorbate concentrations 1.3-fold and was associated with enhanced tolerance to cold stress [122] (Table 3).

### 4.12. NBS-LRR 33 (NL33)

The tomato NBS-LRR 33 (NL33) resistance protein was recently identified based on a candidate gene-based association study with ascorbate concentrations in tomato [123]. The NL33 protein negatively regulates the transcription of ascorbate biosynthetic, ascorbate recycling, and ascorbate-glutathione cycle genes, including *PMM*, *GMP1*, *GMP4*, *GME1*, *GGP1*, *GPP1*, *GPP2*, *L-GalDH*, *L-GalLDH*, *MIOX*, *MDAR*, *DHAR*, and cytosolic *APX* [123]. Knock-down of the *NL33* gene with RNAi in tomato increased ascorbate concentrations 2.7-fold and was associated with enhanced tolerance to methyl viologen-induced stress and gray mold infection (*Botrytis cinerea*) [123] (Table 3). Disrupting the function of the *NL33* gene with genome editing tools represents another transgene-free strategy to increase ascorbate concentrations in crops.

### 4.13. Non-specific Lipid Transfer Protein-1 (nsLTP1)

The potato thermo-tolerance gene *non-specific lipid transfer protein-1* (*nsLTP1*) was first identified in a functional screen of potato high-temperature stress-responsive genes to impart heat tolerance in yeast [124]. The nsLTP1 protein in potato has now recently been identified to positively regulate the transcription of ascorbate-glutathione cycle and stress-related genes, including *APX*, *catalase* (*CAT*), *superoxide dismutase* (*SOD*), *heat shock protein 20* (*HSP20*), *HSP70,* and *HSP90* [125]. Increased expression of the *nsLTP1* gene increased ascorbate concentrations 2.3-fold in potato and was associated with enhanced tolerance to heat, drought, and salt stress [125] (Table 3).

### 4.14. VITAMIN C-3 (VTC3)

The *Arabidopsis vtc3* mutants were first identified in a forward genetic screen for ozone-sensitive mutants and contained 40% of wild-type ascorbate levels [36]. Over a decade later, the *VTC3* gene was mapped to a unique dual-function polypeptide containing an N-terminal protein kinase domain and a C-terminal protein phosphatase domain [126]. The *Arabidopsis vtc3* mutants were reported to be defective in their ability to increase ascorbate concentrations in response to light and heat [126]. Little is known about the mechanism of action for VTC3, but it is proposed to act as a signal transduction protein that can perceive a change in the environment to influence cellular ascorbate levels [126]. It has been suggested that VTC3 may play a role in the *GGP* uORF sensing mechanism of ascorbate levels [44]. Increased expression of the *VTC3* gene in the *Arabidopsis vtc3* mutants recovered ascorbate concentrations back to wild-type levels; however, increased expression of the *VTC3* gene in wild-type *Arabidopsis* did not significantly change ascorbate concentrations [126] (Table 3).

### 4.15. WAX1 

The saltwater cress (*Eutrema salsugineum*) *WAX1* gene was identified based on being rapidly induced in abiotic stress-treated saltwater cress plants [127]. The *WAX1* gene encodes a MYB transcription factor that positively regulates wax related genes, as well as ascorbate biosynthetic genes, including *GGP1*, *L-GalDH*, and *MIOX4* [127]. Increased expression of the *WAX1* gene with the constitutive super-promoter in *Arabidopsis* increased ascorbate concentrations 1.3-fold but severely disrupted plant growth and development [127] (Table 3). Increased expression of the *WAX1* gene with the *Arabidopsis RD29A* stress-inducible promoter, instead of the constitutive super-promoter, increased ascorbate concentrations 1.4-fold under drought stress and was associated with enhanced tolerance to drought stress [127] (Table 3).

### 4.16. Zinc-Finger 3 (ZF3)

The tomato C2H2-type *zinc finger 3* (*ZF3*) gene was first identified based on being induced in salt-stressed tomato plants [128]. The ZF3 protein in tomato has now recently been identified to interact with CSN5B and inhibit the interaction between CSN5B and GMP [129]. By doing so, ZF3 prevents the ubiquitination and degradation of GMP through the 26S proteasome pathway, promoting the accumulation of GMP [129]. Knock-down of the *ZF3* gene with RNAi in tomato reduced ascorbate to 80% of wild-type levels [129]. Increased expression of the tomato *ZF3* gene increased ascorbate concentrations 1.8-fold and 2.1-fold in *Arabidopsis* and tomato, respectively, and was associated with enhanced tolerance to salt stress [129] (Table 3).

## 5. Conclusions and Future Perspectives

Increased expression of single ascorbate biosynthetic genes has been the most utilized and effective strategy to increase ascorbate concentrations in plants and has been associated with enhanced tolerance to multiple abiotic stresses. Increased expression of the *GGP* gene from the L-galactose pathway has consistently generated the largest increases in ascorbate concentrations in a range of species relative to other ascorbate biosynthetic genes. Increased co-expression of the *GGP* gene with other biosynthetic genes from the L-galactose pathway should provide further opportunities to increase ascorbate concentrations in crops to greater than that of the *GGP* gene alone. For example, synergistic effects on ascorbate concentrations were observed when the co-expression of the *GGP* + *GPP* genes or the *GGP* + L-*GalLDH* genes were increased in *Arabidopsis* [24] and the transient co-expression of the *GGP* + *GME* genes were increased in *N. benthamiana* [40]. A recent study that pyramided *GMP1 + GME2 + GGP1 + GPP1* in tomato, however, observed the same fold increase in ascorbate concentrations for that of the *GGP* gene alone. Further research is therefore warranted into increased co-expression of ascorbate biosynthetic genes in plants to better understand precisely which combination of genes leads to the greatest fold increases in ascorbate concentrations.

Increased expression of single ascorbate recycling genes has been the second most common strategy to increase ascorbate concentrations in plants and has also been associated with enhanced tolerance to a broad range of abiotic stress. Increased expression of the *DHAR* gene has consistently generated increased concentrations of reduced ascorbate in a range of species, whereas increased expression of the *MDAR* gene has not always increased reduced ascorbate concentrations. Increased co-expression of the *DHAR* gene with ascorbate biosynthetic genes, such as the *GGP* gene, could present one avenue to increase not only the concentration of total ascorbate, but also to maintain the ascorbate in the reduced, active form for improved stress tolerance.

More recently, the manipulation of ascorbate regulatory factors has emerged as a third strategy to increase ascorbate concentrations in crops. Typically, this strategy has focused on increased expression of a single transcriptional activator of ascorbate biosynthetic genes. However, we propose that more focus should be paid to the discovery and manipulation of transcriptional and translational repressors of ascorbate biosynthetic genes or regulatory factors that target ascorbate biosynthetic proteins for degradation, since they can be targeted for disruption with genome editing tools as an efficient approach to produce transgene-free crops. Genome edited plants offer many advantages over transgenic plants, for example, by avoiding the occurrence of transgene silencing and the random insertion of T-DNAs [130]. Moreover, since plants with genome edited-induced indels (small insertions and deletions) due to errors in non-homologous end joining do not rely on the presence of foreign DNA, they are less likely to be subjected to regulatory oversight. Disrupting the *cis*-acting *GGP* uORF in *Arabidopsis*, lettuce, and tomato with the CRISPR/Cas9 genome editing system was recently reported to significantly increase ascorbate concentrations, as well as improve methyl viologen-induced oxidative stress tolerance in the case of lettuce [117,118]. As the *GGP* uORF is highly conserved from mosses to angiosperms, editing the *GGP* uORF represents a promising transgene-free strategy to increase ascorbate concentrations for enhanced stress tolerance in most crops [115,116]. Other ascorbate regulatory factors that could be targeted with genome editing include *AMR1*, *CSN5B*, *CSN8*, and *NL33,* which have all been reported to increase ascorbate concentrations and enhance stress tolerance when disrupted with T-DNA insertions or downregulated with RNAi [109,112,123].

Finally, it is worth noting that increasing ascorbate concentrations in crops does not come without its challenges [131]. For example, increased expression of ascorbate biosynthetic and recycling genes has been reported to reduce guard cell responsiveness [132], disturb embryo development [133], and induce parthenocarpic (seedless) fruit in tomato [41]. A fine balance must therefore be struck when increasing ascorbate concentrations in crops in order to minimize any deleterious effects on plant growth and development that may arise due to perturbed ascorbate concentrations. Generating mutant alleles of varying strength in the *GGP* uORF with the CRISPR/Cas9 genome editing system represents an attractive strategy to fine-tune ascorbate concentrations in crops for enhanced stress tolerance, whilst minimizing any pleotropic effects on plant growth and development [117,118,130].

## Figures and Tables

**Figure 1 ijms-21-01790-f001:**
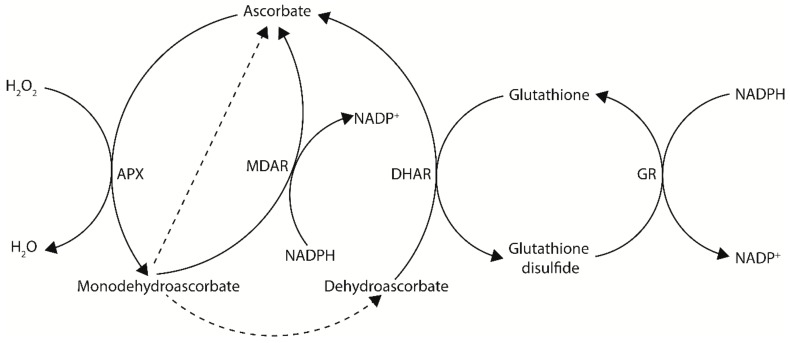
The ascorbate–glutathione cycle—a major antioxidant system of plants cells. In this cycle, electrons flow from NADPH to H_2_O_2_. Dashed arrows indicate non-enzymatic disproportionation. APX, ascorbate peroxidase; MDAR, monodehydroascorbate reductase; DHAR, dehydroascorbate reductase; GR, glutathione reductase.

**Figure 2 ijms-21-01790-f002:**
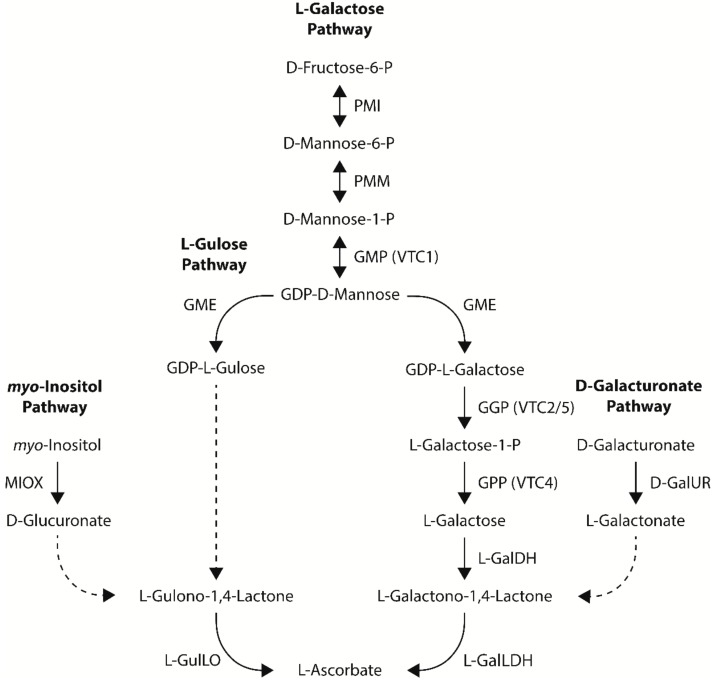
The four proposed ascorbate biosynthetic pathways in higher plants—the L-galactose, L-gulose, *myo*-inositol, and D-galacturonate pathways. Dashed arrows indicate multiple and/or unknown biosynthetic steps in higher plants. PMI, phosphomannose isomerase; PMM, phosphomannose mutase; GMP, GDP-D-mannose pyrophosphorylase; GME, GDP-D-mannose-3′,5′-epimerase; GGP, GDP-L-galactose phosphorylase; GPP, L-galactose-1-P phosphatase; L-GalDH, L-galactose dehydrogenase; L-GalLDH, L-GalLDH L-galactono-1,4-lactone dehydrogenase; L-GulLO, L-gulono-1,4-lactone oxidase; MIOX, *myo*-inositol oxygenase; D-GalUR, D-galacturonate reductase.

**Table 1 ijms-21-01790-t001:** Overview of metabolic engineering strategies utilizing ascorbate biosynthetic genes to increase ascorbate concentrations in model and crop species.

Species Transformed	Gene(s) Transformed	Gene Source(s)	Promoter	Max Fold-Change	Tissue Examined	Stress Tolerance	Reference
L-galactose pathway genes							
Tobacco	*PMM*	Acerola	*CaMV 35S*	~2.5	Leaves	-	[20]
*Arabidopsis*	*PMM-GFP*	*Arabidopsis*	*CaMV 35S*	1.3	Leaves	Methyl viologen	[19]
*Arabidopsis*	*GMP*	*Arabidopsis*	*CaMV 35S*	1.3	Leaves	-	[24]
*Arabidopsis*	*GMP*	*Arabidopsis*	*CaMV 35S*	~1.2^ns^	Seedlings	-	[30]
Rice	*GMP*	*Arabidopsis*	*ZmUbi*	~1.5	Leaves	-	[28]
Tobacco	*GMP*	Peach	*CaMV 35S*	~1.3^ns^	Leaves	-	[29]
Tobacco	*GMP*	Acerola	*MgGMP*	~2.5	Leaves	-	[25]
Tobacco	*GMP*	Acerola	*CaMV 35S*	~2.0	Leaves	-	[25]
Tomato	*GMP1*	Tomato	*CaMV 35S*	~1.4, ~1.2	Leaves, Red Fruit	-	[26]
Tobacco	*GMP + GME*	Peach	*CaMV 35S*	~1.4^ns^, ~1.8^ns^, ~1.0^ns^, ~1.1^ns^	Young Leaves, Old Leaves, Flower Buds, Immature Fruits	-	[29]
Tomato	*GMP1 + GME2*	Tomato	*CaMV 35S*	2.0, ~1.3	Leaves, Red Fruit	Methyl viologen	[26]
Tomato	*GMP1 + GME2 + GGP1 + GPP1*	Tomato	*CaMV 35S*	2.0, ~1.3	Leaves, Red Fruit	Methyl viologen	[26]
*Arabidopsis*	*GME*	*Arabidopsis*	*CaMV 35S*	1.4	Leaves	-	[24]
*Arabidopsis*	*GME*	Alfalfa	*CaMV 35S*	1.8	Leaves	Low pH, drought, salt	[33]
*Arabidopsis*	*GME*	Rose	*CaMV 35S*	1.9	Leaves	-	[34]
Rice	*GME*	*Arabidopsis*	*ZmUbi*	~1.9	Leaves	Salt	[28]
Tobacco	*GME*	Peach	*CaMV 35S*	~1.3^ns^	Leaves	-	[29]
Tomato	*GME1*	Tomato	*CaMV 35S*	1.4, 1.6	Leaves, Fruits	Methyl viologen, cold, salt	[35]
Tomato	*GME2*	Tomato	*CaMV 35S*	~1.8, ~1.2	Leaves, Red Fruit	-	[26]
Tomato	*GME2*	Tomato	*CaMV 35S*	1.3, 1.2	Leaves, Fruits	Methyl viologen, cold, salt	[35]
*Arabidopsis*	*GGP*	Kiwifruit	*CaMV 35S*	4.1	Leaves	-	[40]
Rice	*GGP*	Kiwifruit	*OsLP2*	2.5	Leaves	Salt, ozone	[43]
Strawberry	*GGP*	Kiwifruit	*CaMV 35S*	1.8, 2.1	Leaves, Fruit	-	[41]
Tomato	*GGP*	Kiwifruit	*CaMV 35S*	2.0, 6.2	Leaves, Fruit	-	[41]
*Arabidopsis*	*GGP1*	*Arabidopsis*	*CaMV 35S*	2.9	Leaves	-	[24]
Potato	*GGP1*	*Arabidopsis*	*CaMV 35S*	1.7^ns^	Tubers	-	[41]
Potato	*GGP1*	Potato	*CaMV 35S*	1.8^ns^	Tubers	-	[41]
Potato	*GGP1*	Potato	*StPAT*	3.0	Tubers	-	[41]
Potato	*GGP1*	*Arabidopsis*	*CaMV 35S*	1.5^ns^	Tubers	-	[41]
Rice	*GGP1*	*Arabidopsis*	*CaMV 35S*	2.6	Leaves	Salt	[28]
Tobacco	*GGP1*	Tomato	*CaMV 35S*	1.4	Leaves	Cold	[42]
Tomato	*GGP1*	Tomato	*CaMV 35S*	2.0, ~1.1^ns^	Leaves, Red Fruit	-	[26]
*Arabidopsis*	*GGP1 + L-GalLDH*	*Arabidopsis*	*CaMV 35S*	3.6	Leaves	-	[24]
*Arabidopsis*	*GGP1 + GPP*	*Arabidopsis*	*CaMV 35S*	4.1	Leaves	-	[24]
Tomato	*GGP1 + GPP1*	Tomato	*CaMV 35S*	~1.8, ~1.0^ns^	Leaves, Red Fruit	Methyl viologen	[26]
Potato	*GGP2*	Potato	*CaMV 35S*	2.4	Tubers	-	[41]
Potato	*GGP2*	Potato	*StPAT*	3.1	Tubers	-	[41]
*Arabidopsis*	*GPP*	*Arabidopsis*	*CaMV 35S*	1.5	Leaves	-	[24]
Rice	*GPP*	*Arabidopsis*	*ZmUbi*	~1.4	Leaves	-	[28]
Tomato	*GPP1*	Tomato	*CaMV 35S*	~1.7, ~1.1^ns^	Leaves, Red Fruit	-	[26]
*Arabidopsis*	*L-GalDH*	*Arabidopsis*	*CaMV 35S*	1.2	Leaves	-	[24]
Rice	*L-GalDH*	*Arabidopsis*	*ZmUbi*	~1.7	Leaves	-	[28]
Tobacco	*L-GalDH*	*Arabidopsis*	*CaMV 35S*	~1.1^ns^	Leaves	-	[46]
*Arabidopsis*	*L-GalLDH*	*Arabidopsis*	*CaMV 35S*	1.8	Leaves	-	[24]
Lettuce	*L-GalLDH*	Lettuce	*PspetE*	1.3	Leaves	-	[48]
Lettuce	*L-GalLDH*	*Arabidopsis*	*CaMV 35S*	3.2	Leaves	-	[49]
Lily	*L-GalLDH*	Apple	*CaMV 35S*	7.0	Leaves	-	[50]
Rice	*L-GalLDH*	Rapeseed	*ZmUbi*	~1.5	Leaves	-	[28]
Rice	*L-GalLDH*	Rice	*ZmUbi*	1.3	Leaves	-	[51]
Tobacco	*L-GalLDH*	Rose	*CaMV 35S*	2.1	Leaves	Salt, methyl viologen	[52]
Tobacco	*L-GalLDH*	Sweet potato	*CaMV 35S*	1.0^ns^	Leaves	-	[53]
L-gulose and *myo*-inositol pathway genes							
*Arabidopsis*	*L-GulLO3*	*Arabidopsis*	*CaMV 35S*	~0.8^ns^	Leaves	-	[61]
*Arabidopsis*	*L-GulLO5*	*Arabidopsis*	*CaMV 35S*	~1.1^ns^	Leaves	-	[61]
Rice	*MIOX*	Rice	*CaMV 35S*	1.0^ns^	Leaves	Drought	[68]
Tomato	*MIOX2*	*Arabidopsis*	*CaMV 35S*	0.5, 1.4, 1.3	Leaves, Green Fruit, Red Fruit	-	[27]
*Arabidopsis*	*MIOX4*	*Arabidopsis*	*CaMV 35S*	1.6	Leaves	Salt, cold, heat, pyrene	[56]
*Arabidopsis*	*MIOX4*	*Arabidopsis*	*CaMV 35S*	3.0	Leaves	-	[63]
*Arabidopsis*	*MIOX4*	*Arabidopsis*	*CaMV 35S*	1.7	Leaves	Heat	[64]
*Arabidopsis*	*MIOX4*	*Arabidopsis*	*CaMV 35S*	1.0^ns^	Leaves	-	[67]
D-galacturonate pathway genes							
*Arabidopsis*	*D-GalUR*	Strawberry	*CaMV 35S*	3.0	Leaves	-	[72]
Potato	*D-GalUR*	Strawberry	*CaMV 35S*	2.0	Tubers	Methyl viologen, salt, drought	[73]
Potato	*D-GalUR*	Strawberry	*CaMV 35S*	~2.1	Tubers	Methyl viologen, salt, zinc chloride	[75]
Potato	*D-GalUR*	Strawberry	*CaMV 35S*	~1.7	Tubers	Salt	[74]
Tomato	*D-GalUR*	Strawberry	*CaMV 35S*	2.5	Fruit	Methyl viologen, salt, drought	[60]
Tomato	*D-GalUR*	Strawberry	*CaMV 35S*	1.8, 2.0	Leaves, Red Fruit	Methyl viologen, salt, cold	[77]
Tomato	*D-GalUR*	Strawberry	*CaMV 35S*	1.3, ~1.0^ns^, 1.4	Leaves, Green Fruit, Light Red Fruit	-	[76]
Tomato	*D-GalUR*	Strawberry	*SlPG*	~1.0^ns^, ~1.0^ns^, 1.4	Leaves, Green Fruit, Light Red Fruit	-	[76]
Animal and yeast genes							
Tobacco	*ALO*	Yeast	*CaMV 35S*	~2.5	Leaves	Cold	[79]
Tobacco	*ALO*	Yeast	*CaMV 35S*	~1.5, ~1.5, ~2.0	Young Leaves, Mature Leaves, Old Leaves	Methyl viologen, high light, Al toxicity	[80]
Tomato	*ALO*	Yeast	*CaMV 35S*	1.5, 1.3, 1.1^ns^	Leaves, Green Fruit, Red Fruit	-	[27]
Stylo	*ALO + NCED*	Yeast/stylo	*CaMV 35S*	3.4	Leaves	Drought, cold	[79]
Tobacco	*ALO + NCED*	Yeast/stylo	*CaMV 35S*	~2.5	Leaves	Drought, cold	[79]
Tomato	*GMP*	Yeast	*CaMV 35S*	1.7, 1.5, 1.4	Leaves, Green Fruit, Red Fruit	-	[27]
*Arabidopsis*	*L-GulLO*	Rat	*CaMV 35S*	1.8	Leaves	Salt, cold, heat, pyrene	[56]
*Arabidopsis*	*L-GulLO*	Rat	*CaMV 35S*	2.0	Leaves	-	[57]
Lettuce	*L-GulLO*	Rat	*CaMV 35S*	7.0	Leaves	-	[58]
Potato	*L-GulLO*	Rat	*CaMV 35S*	2.4	Tubers	Methyl viologen, salt, drought	[59]
Tobacco	*L-GulLO*	Rat	*CaMV 35S*	7.0	Leaves	-	[58]
Tomato	*L-GulLO*	Rat	*CaMV 35S*	1.5	Red fruit	Methyl viologen, salt, drought	[60]

PMM, phosphomannose mutase; GFP, green fluorescent protein; GMP, GDP-D-mannose pyrophosphorylase; GME, GDP-D-mannose-3′,5′-epimerase; GGP, GDP-L-galactose phosphorylase; GPP, L-galactose-1-P phosphatase; L-GalDH, L-galactose dehydrogenase; L-GalLDH, L-GalLDH L-galactono-1,4-lactone dehydrogenase; L-GulLO, L-gulono-1,4-lactone oxidase; MIOX, *myo*-inositol oxygenase; D-GalUR, D-galacturonate reductase; ALO, D-arabinono-1,4-lactone oxidase; NCED, 9-cis-epoxycarotenoid dioxygenase; CaMV 35S, cauliflower mosaic virus 35S constitutive promoter; ZmUbi, maize ubiquitin constitutive promoter; MgGMP, native promoter of acerola GMP; OsLP2, rice leaf panicle 2 leaf-specific promoter; StPAT, potato polyubiquitin constitutive promoter; PspetE, pea plastocyanin constitutive promoter; SlPG, tomato polygalucturonase fruit-specific promoter; ~, approximately; ns, non-significant.

**Table 2 ijms-21-01790-t002:** Overview of metabolic engineering strategies utilizing ascorbate recycling genes to increase ascorbate concentrations in model and crop species.

Species Transformed	Gene(s) Transformed	Gene Source(s)	Promoter	Max Fold-Change	Tissue Examined	Stress Tolerance	Reference
Tobacco	*MDAR*	Acerola	*CaMV 35S*	2.0	Leaves	Salt	[83]
Tobacco	*MDAR*	Mangrove	*CaMV 35S*	~1.3^ns^	Leaves	Salt	[87]
Tomato	*MDAR*	Tomato	*CaMV 35S*	1.2	Leaves	Cold, heat, methyl viologen	[86]
Tomato	*MDAR*	Tomato	*FMV 34S*	1.2^ns^, 1.0^ns^	Leaves, red fruit	-	[88]
Tobacco	*MDAR1*	*Arabidopsis*	*CaMV 35S*	2.2	Leaves	Ozone, salt, drought	[84]
Tobacco	*MDAR1*	*Arabidopsis*	*CaMV 35S*	~1.2	Roots	-	[85]
*Arabidopsis*	*DHAR*	Chinese tulip tree	*CaMV 35S*	~1.4	Leaves	Salt, drought	[90]
*Arabidopsis*	*DHAR*	Rose	*CaMV 35S*	3.0	Leaves	-	[34]
Maize	*DHAR*	Wheat	*ZmUbi*	1.8, 1.9	Leaves, kernels	-	[95]
Potato	*DHAR*	Sesame	*CaMV 35S*	1.5, 1.6	Leaves, tubers	-	[97]
Potato	*DHAR*	Sesame	*PtPal*	1.3	Tubers	-	[97]
Tobacco	*DHAR*	Wheat	*CaMV 35S*	2.1	Leaves	Ozone	[103]
Tobacco	*DHAR*	Human	*CaMV 35S*	1.6	Leaves	Methyl viologen, H_2_O_2_, cold, salt	[102]
Tobacco	*DHAR*	Human	*CaMV 35S*	1.6, 2.0	Young leaves, mature leaves	Methyl viologen	[101]
Tobacco	*DHAR*	Wheat	*CaMV 35S*	2.4, 3.9, 2.2	Expanding leaves, mature leaves, presenescent leaves	-	[95]
Tomato	*DHAR*	Tomato	*FMV 34S*	1.1^ns^, 1.6, 1.6	Leaves, green fruit, red fruit	-	[88]
Tobacco	*DHAR + CuZnSOD + APX*	Human + pea + pea	*CaMV 35S*	1.5	Leaves	Methyl viologen, salt	[104]
*Arabidopsis*	*DHAR1*	*Arabidopsis*	*CaMV 35S*	3.3	Leaves	High light, heat, paraquat	[91]
*Arabidopsis*	*DHAR1*	Kiwifruit	*CaMV 35S*	~1.5	Leaves	Salt	[92]
*Arabidopsis*	*DHAR1*	Rice	*CaMV 35S*	1.2	Leaves	Salt	[93]
Maize	*DHAR1*	Rice	*HvHor*	6.1	Kernels	-	[96]
Potato	*DHAR1*	*Arabidopsis*	*CaMV 35S*	2.8	Leaves	Methyl viologen, drought, salt	[98]
Potato	*DHAR1*	Potato	*CaMV 35S*	1.7, 1.3	Leaves, tubers	-	[99]
Rice	*DHAR1*	Rice	*ZmUbi*	~1.7	Leaves	-	[100]
Tobacco	*DHAR1*	Rice	*PsPrrn*	1.6	Leaves	Salt, cold	[106]
Tomato	*DHAR1*	Potato	*CaMV 35S*	1.9, 1.4	Leaves, fruit	Methyl viologen, salt	[107]
Tobacco	*DHAR1 + GR*	Rice + *E. coli*	*PsPrrn*	~2.5	Leaves	Salt, cold, methyl viologen	[106]
*Arabidopsis*	*DHAR2*	Kiwifruit	*CaMV 35S*	~1.4	Leaves	Salt	[109]
Potato	*DHAR2*	Potato	*CaMV 35S*	1.5, ~1.1^ns^	Leaves, tubers	-	[99]
Tobacco	*DHAR2*	*Arabidopsis*	*CaMV 35S*	2.1	Leaves	Ozone, drought, salt	[105]
Tobacco	*DHAR2*	*Arabidopsis*	*CaMV 35S*	~1.3	Roots	Aluminium	[85]
Tomato	*DHAR2*	Potato	*CaMV 35S*	1.8, ~1.1^ns^	Leaves, fruit	Methyl viologen, salt	[107]
Tomato	*DHAR2*	Pear	*CaMV 35S*	1.5	Leaves	Salt, cold	[108]
*Arabidopsis*	*DHAR3*	Sweet potato	*CaMV 35S*	~1.1^ns^	Leaves	Salt, drought	[94]

MDAR, monodehydroascorbate reductase; DHAR, dehydroascorbate reductase; CuZnSOD, copper zinc superoxide dismutase; APX, ascorbate peroxidase; GR, glutathione reductase; E. coli, Escherichia coli; CaMV 35S, cauliflower mosaic virus 35S constitutive promoter; FMV 34S, figwort mosaic virus 34S constitutive promoter; ZmUbi, maize ubiquitin constitutive promoter; PtPal, potato patatin tuber-specific promoter; HvHor, barley D-hordein endosperm-specific promoter; PsPrrn, pea plastid rRNA operon constitutive promoter; ~, approximately; ns, non-significant.

**Table 3 ijms-21-01790-t003:** Overview of metabolic engineering strategies utilizing ascorbate regulatory factors to increase ascorbate concentrations in model and crop species.

Species	Regulatory Factor	Strategy	Gene Source	Promoter	Max Fold-Change	Tissue Examined	Stress Tolerance	Reference
*Arabidopsis*	*AMR1*	T-DNA insertion	-	-	2.0	Leaves	Ozone	[109]
Tomato	*bHLH59*	Increased expression	Tomato	*CaMV 35S*	~1.5	Fruit	Methyl viologen	[110]
*Arabidopsis*	*CSN5B*	T-DNA insertion	-	-	~1.4	Seedlings	Salt	[112]
*Arabidopsis*	*CSN8*	T-DNA insertion	-	-	~1.8	Seedlings	-	[112]
Tomato	*Dof22*	RNAi	-	-	1.3, 1.6	Leaves, red fruit	-	[113]
*Arabidopsis*	*ERF98*	Increased expression	*Arabidopsis*	*CaMV 35S*	1.7	Leaves	Salt	[114]
*Arabidopsis*	*GGP1* uORF	Genome editing	-	*-*	1.7	Leaves	-	[117]
Lettuce	*GGP1* uORF	Genome editing	-	*-*	1.4	Leaves	Methyl viologen	[117]
Lettuce	*GGP2* uORF	Genome editing	-	*-*	2.6	Leaves	Methyl viologen	[117]
Tomato	*GGP2* uORF	Genome editing	-	*-*	~1.4	Leaves	-	[118]
Tomato	*HZ24*	Increased expression	Tomato	*CaMV 35S*	1.5, ~1.2	Leaves, breaker fruit	Methyl viologen	[119]
*Arabidopsis*	*KJC1*	Increased expression	*Arabidopsis*	*CaMV 35S*	1.4	Seedlings	-	[30]
*Arabidopsis*	*KJC1 + GMP*	Increased expression	*Arabidopsis*	*CaMV 35S*	~1.0^ns^	Seedlings	-	[30]
*Arabidopsis*	*KJC2*	Increased expression	*Arabidopsis*	*CaMV 35S*	~1.0^ns^	Seedlings	-	[30]
Tobacco	*MYB5*	Increased expression	Pear	*CaMV 35S*	~1.3	Leaves	Cold	[122]
Tomato	*NL33*	RNAi	-	*-*	2.7, 1.3	Leaves, red fruit	Methyl viologen, *Botrytis cinerea*	[123]
Potato	*nsLTP1*	Increased expression	Potato	*CaMV 35S*	2.3	Leaves	Heat, drought, salt	[125]
*Arabidopsis*	*VTC3*	Increased expression	*Arabidopsis*	*CaMV 35S*	~0.8^ns^	Seedlings	-	[126]
*Arabidopsis*	*WAX1*	Increased expression	Saltwater cress	*SP*	1.3	Leaves	-	[127]
*Arabidopsis*	*WAX1*	Increased expression	Saltwater cress	*AtRD29A*	~1.4*	Leaves	Drought	[127]
*Arabidopsis*	*ZF3*	Increased expression	Tomato	*CaMV 35S*	1.8	Leaves	Salt	[129]
Tomato	*ZF3*	Increased expression	Tomato	*CaMV 35S*	~2.1	Leaves	Salt	[129]

AMR1, ascorbic acid mannose pathway regulator 1; CSN5B, COP9 signalosome subunit 5B; Dof22, DNA-binding with one finger 22; ERF98, ethylene response factor 98; GGP uORF, GDP-L-galactose phosphorylase upstream open reading frame; HZ24, HD-Zip I transcription factor 24; KJC, KONJAC; MYB5, myeloblastosis transcription factor 5; NL33, NBS-LRR 33; nsLTP1, non-specific lipid transfer protein-1; VTC3, VITAMIN C-3; CaMV 35S, cauliflower mosaic virus 35S constitutive promoter; SP, super-promoter constitutive promoter; AtRD29A, Arabidopsis stress-inducible promoter; ~, approximately; ns, non-significant; *under drought stress.

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
