# Peer review of "Manipulation of Ascorbate Biosynthetic, Recycling, and Regulatory Pathways for Improved Abiotic Stress Tolerance in Plants"

_ijms, 2020, doi:10.3390/ijms21051790_

Round 1

Reviewer 1 Report

This interesting and useful review discusses the pathways of ascorbate biosynthesis, recycling, and regulation in plants, within the perspective of metabolic engineering strategies to increase ascorbate accumulation in crops. I found the discussion of the molecular factors that regulate ascorbate synthesis particularly well presented and insightful. In the conclusions section, the author consider strategies to increase ascorbate accumulation in plants and propose that more research focus is placed on the discovery and manipulation of transcriptional and translational repressors of ascorbate synthesis genes or regulatory factors that target these proteins for degradation.

In lines 337-338, the authors conclude that “DHAR is the rate-limiting step in ascorbate recycling in plants relative to MDAR” However, this conclusion is perhaps questionable given that studies on the Arabidopsis dhar mutants have shown that DHAR activities can be decreased substantially without marked effects on ascorbate pools (Rahantaniaina et al., 2017. Plant Physiol 174: 956–971).

I wonder if the authors might consider the possible negative impacts of enhancing ascorbate concentrations in plants. They state that ascorbate accumulation is associated with enhanced tolerance to multiple abiotic stresses but what about biotic stresses and effects on ROS signalling. In this regard, it is interesting that in the Introduction, the authors place emphasis on the traditional concept of ROS toxicity in the text, as in lines 47-49 “ If left unchecked, the build-up of ROS can cause damage to proteins, lipids, DNA, and RNA, and ultimately lead to cell death”. The authors cite an earlier review in this regard: Sharma et al.2012. However, this reference has several inaccuracies not least in their estimates of ROS production in plants. The context with which ascorbate is discussed might be somewhat biased by this rather outdated notion. It would have been perhaps more interesting if the prospects for producing plants with high levels of ascorbate was discussed within the wider context of impacts on redox signalling and interacting phytohormone pathways.

Author Response

Agreed. We have removed the conclusion that DHAR is the rate-limiting step in ascorbate recycling (tracked, Lines 353-354).

Agreed. There are indeed many challenges facing increased ascorbate concentrations plants, with inadvertent effects on plant growth and development reported in the literature. A paragraph highlighting this has been added to the end of the Conclusions and Future Perspectives: “Finally, it is worth noting that increasing ascorbate concentrations in crops does not come without its challenges [133]. For example, increased expression of ascorbate biosynthetic and recycling genes has been reported to reduce guard cell responsiveness [134], disturb embryo development [135], and induce parthenocarpic (seedless) fruit in tomato [42]. A fine balance must therefore be struck when increasing ascorbate concentrations in crops in order to minimize any deleterious effects on plant growth and development that may arise due to perturbed ascorbate concentrations. Generating mutant alleles of varying strength in the GGP uORF with the CRISPR/Cas9 genome editing system represents an attractive strategy to fine-tune ascorbate concentrations in crops for enhanced stress tolerance, whilst minimizing any pleotropic effects on plant growth and development [119, 120, 132].” (tracked, Lines 582-591). Moreover, references to Sharma et al., (2012) have been removed and the introduction updated (tracked, Lines 46-49).

Reviewer 2 Report

Ronan C. Broad and co-workers present the manuscript “Manipulation of Ascorbate Biosynthetic, Recycling, and Regulatory Pathways for Improved Stress Tolerance in Plants”, where they review ascorbate genetic pathways including primary strategies to increase ascorbate content in crop species (genome editing).

The content   is within the scope of Antioxidants.

However, after reading of the manuscript I think that some further improvements are possible before its publication in the journal. Below please find my comments.

English throughout the whole manuscript should be revised.

In the title is necessary to insert the word “Abiotic” because the review focalizes that kind of stress

Lines 14 and 553.  Do the authors mean “if not all”.  The sentences are not clear. Please rephrase and specify.

Lines 52-54  “In plants, ascorbate is the most abundant  water-soluble antioxidant and is found in high concentrations in the cytosol, peroxisomes, nuclei, mitochondria, and chloroplasts, [9]”.  Some information  should be provided regarding this different concentrations, including also the apoplastic ascorbate concentration. It is preferible add a Table or colored Figure (plant cell with various districts and concentrations)

Lines 81. “The genes encoding the enzymatic steps are shown in italics”. ??? The genes or the enzymes?.  In Figure are shown the acronyms of enzyme names!!! Please correct

Table 1 and Table2. Write and report the words “transformed” and “Arabidopsis” and “fold-change” on the same line

Scientific name of organisms should be checked thoroughly in the entire manuscript.

If possible, it is invite the authors to make colored the reported Figures.

Author Response

The manuscript has been carefully revised and minor typographical errors have been corrected throughout the manuscript (tracked, many lines).

Agreed. The word abiotic has been added to the title (tracked, Line 3).

Clarified. The phrase ‘if not all’ has been removed from the manuscript (tracked, Line 14 and 578).

Agreed. The subcellular concentrations of ascorbate have now been provided, including a brief description of the vacuolar and apoplastic concentrations (tracked, Lines 54-57). The subcellular concentrations of ascorbate have been reviewed well in ‘Zechmann, B., Subcellular distribution of ascorbate in plants. Plant Signal Behav 2011, 6, (3), 360-363.’, including a figure of subcellular concentrations, therefore we do not feel an additional table or figure is warranted within our manuscript.

Clarified. Figures 1 and 2 have been updated so that the enzyme names are presented (not italicized). The captions of Figures 1 and 2 have also been updated accordingly (tracked, Line 86 and 309-310).

This issue arose due to formatting following submission of the manuscript. Tables 1, 2, and 3 have all been re-formatted to better fit the text in the columns. 

The scientific names of organisms have been carefully checked throughout the manuscript and a typo was found in Solanum lycopersicum L. (tracked, Line 137) and an L. omitted from Rattus rattus L. (tracked, Line 239). Both have been corrected.

While we appreciate the suggestion to add color to the figures and have previously considered it ourselves, we have decided to keep the figures simply black and white for ease of printing, enhanced contrast, and to benefit people with color blindness.